# Neurosymbolic Deep Generative Models for Sequence Data with Relational Constraints

**Halley Young**
Department of Computer Science
University of Pennsylvania
`halleyy@seas.upenn.edu`

**Maxwell Du**
Department of Computer Science
University of Pennsylvania
`mdu@seas.upenn.edu`

**Osbert Bastani**
Department of Computer Science
University of Pennsylvania
obastani@seas.upenn.edu

## Abstract

There has been significant recent progress designing deep generative models that generate realistic sequence data such as text or music. Nevertheless, it remains difficult to incorporate high-level structure to guide the generative process, and many such models perform well on local coherence, but less so on global coherence. We propose a novel approach for incorporating global structure in the form of relational constraints between different subcomponents of an example (e.g., lines of a poem or measures of music). Our generative model has two parts: (i) one model to generate a realistic set of relational constraints, and (ii) a second model to generate realistic data satisfying these constraints. For model (i), we propose a program synthesis algorithm that infers the relational constraints present in the training data, and then learn a generative model based on the resulting constraint data. In our experiments, we show that our approach significantly improves over state-of-the-art in terms of capturing high-level structure in the data, while performing comparably or better in terms of low-level structure.

## 1 Introduction

There has been tremendous recent progress in designing deep generative models for generating sequence data such as natural language (Vaswani et al., 2017) or music (Huang et al., 2019). These approaches leverage the vast quantities of data available in conjunction with unsupervised and self-supervised learning to learn probabilistic models of the data; then, new examples can be generated by sampling from these models, with the possibility of conditioning on initial elements of the sequence.

A key challenge facing deep generative models is the difficulty incorporating high-level structure into the generated examples—e.g., rhyming and meter across lines of a poem, or repetition across measures of a piece of music. Capturing high-level structure is important for improving the quality of the generated data, especially in low-data regimes where only small numbers of examples are available; intuitively, knowledge of the structure compresses the amount of information the generative model has to learn. Furthermore, *explicit* representations of structure (i.e., symbolically rather than as a vector embedding) has the benefit that users can modify the structure to guide generation.

Recently, Young et al. (2019) proposed *neurosymbolic generative models* for incorporating high-level structure into image generation, focusing on simple 2D repeating patterns in images of building facades (e.g., repeating windows). The basic idea is to leverage *program synthesis* to extract structure from data—in particular, given an example image $x$, they devise an algorithm $\mathcal{A}$ that extracts a

36th Conference on Neural Information Processing Systems (NeurIPS 2022).

program $c = \mathcal{A}(x)$ that represents the set of 2D repeating patterns present in training examples $x$. Then, using the pairs $(x, c)$, they train two generative models: (i) a model $p_\phi(c)$ that generates a program, and (ii) a model $p_\theta(x \mid c)$ that generates an image that contains the structure represented by $c$. However, their approach is heavily tailored to images in two ways. First, their representation of structure is geared towards simple patterns occurring in images of building facades. Second, their algorithm $\mathcal{A}$ is specifically designed to extract this kind of program from an image, as are their models $p_\phi(c)$ for generating programs $c$ and $p_\theta(x \mid c)$ for generating images $x$ conditioned on $c$.

We represent the relational constraints $c_x$ present in an example $x$ by relating each subcomponent $w$ of a given example $x$ with a *prototype* $\tilde{w}$, which can be thought of as the "original" subcomponent from which $w$ is constructed. In particular, the relationship between $w$ and $\tilde{w}$ is labeled with a set of relations $R$, which encodes the constraint that $w$ and $\tilde{w}$ should satisfy relation $r$ for each $r \in R$. Importantly, while each subcomponent is associated with a single prototype, each prototype may be associated with multiple subcomponents. As a consequence, different subcomponents associated with the same prototype are related in some way. This representation is compact, only requiring linearly many constraints in the number of subcomponents in $x$ (assuming the number of prototypes is constant). Compactness ensures the representation both generalizes well and is easy to generate.

Then, we design a synthesis algorithm that can extract an optimal representation of the structure present in a training example $x$ (i.e., the relational constraints $c_x$). We show how to express the synthesis problem as a constrained combinatorial optimization problem, which we solve using an SMT solver Z3 (De Moura & Bjørner, 2008). Next, we represent relational constraints $c$ as sequences, and design the model $p_\phi(c)$ to be a specialized sequence VAE. Finally, we propose three possible designs of $p_\theta(x \mid c)$, all of which try to identify an example $x$ that is realistic (e.g., according to a pretrained model $p_\theta(x)$) while simultaneously satisfies the given constraints $c$.

We evaluate our approach on two tasks: poetry generation, where the relational constraints include rhyming lines or lines with shared meter, and music generation, where the relational constraints include equality in terms of pitch or rhythm, that one measure is a transposition of another (i.e., pitches shifted up or down by a constant amount), etc. We show that our approaches outperform or perform similarly to state-of-the-art models in terms of low-level structure, while significantly outperforming them in terms of high-level structure. We also perform a user study in the poetry domain to determine user-perceived quality of the generated poetry along three dimensions (structure, lyricism, and coherence), and found that on average, our approach outperformed state-of-the-art baselines including GPT-2. Finally, we demonstrate how our approach allows users to guide the generation process without sacrificing overall realism by specifying values of constraints.

**Example.** Figure 1 illustrates how our approach is applied to generate poetry. During training, our approach uses program synthesis to infer relational constraints $c_x$ present in the examples $x$, and uses both $x$ and $c_x$ to train the generative models. Here, $c_x$ is a bipartite graph, where the LHS vertices are *prototypes*, and the RHS vertices correspond to lines of $x$. Each vertex on the right is connected to exactly one prototype, and is labeled with constraints on how it should relate to its prototype. To generate new examples, it first samples relational constraints $c$, and then samples an example $x$ that satisfies $c$—i.e., we need to choose a line to fill each RHS node in a way that the line satisfies the relations with its prototype. Furthermore, a user can modify the sampled constraint $c$ to guide the generative process. Thus, our approach enables users to flexibly incorporate domain knowledge on the high-level structure of the data into the generative process, both in terms of the relational constraints included and by allowing them to modify the generated relational constraints.

**Related work.** There has been recent work using program synthesis to improve machine learning. For instance, it has been applied to unsupervised learning of latent structure in drawings (Ellis et al., 2015) and to reinforcement learning (Verma et al., 2018). These techniques have benefits such as improving interpretability (Verma et al., 2018; Ellis et al., 2020), enabling learning from fewer examples (Ellis et al., 2015), generalizing more robustly (Inala et al., 2019), and being easier to formally verify (Bastani et al., 2018). More recently, there has been work leveraging program synthesis in conjunction with deep learning, where the DNN handles perception and program synthesis handles high-level structure (Ellis et al., 2017), including work in the lifelong learning setting (Valkov et al., 2018). In contrast to these approaches, our focus is on generative models. In particular, we extend recent work leveraging these ideas for image generation to incorporating high-level relational structure into sequence generation tasks (Young et al., 2019). Finally, much research over the past

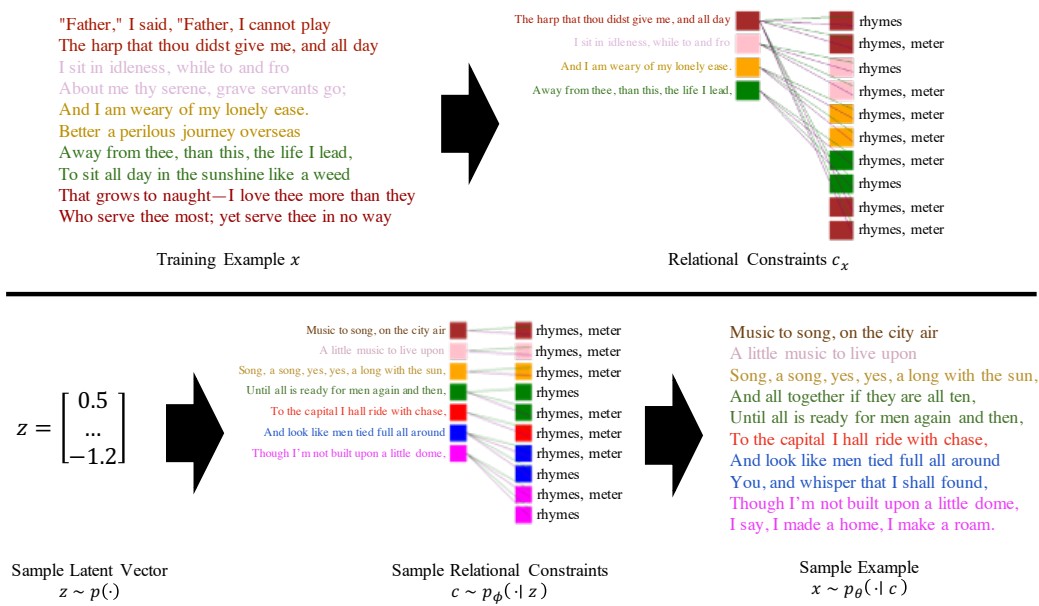

Figure 1: Top: For each training example $x$, our algorithm uses program synthesis to infer the relational constraints $c_x = \mathcal{A}(x)$ present in $x$. Then, it uses $c_x$ and $x$ to train $p_\phi(c_x)$ and $p_\theta(x \mid c_x)$. Bottom: Process for generating a sample $x$ from the learned VAE $p_\phi(c_x \mid z)$ (where $z$ is Gaussian noise) and model $p_\theta(x \mid c_x)$. Lines with the same prototype are shown in the same color; metrical constraints are represented as purple and rhyme constraints as green edges.

few decades has focused on music and poetry generation, and on using relational constraints in neural models; we include a discussion of the most relevant such research in Appendix A.

## 2 Background on Neurosymbolic Generative Models

Consider the problem of learning a generative model given training data from the underlying distribution. Given training examples $x_1, ..., x_k \sim p^*$, our goal is to learn a generative model $p_\theta \approx p^*$ from which we can draw additional samples $x \sim p_\theta$. We consider sequence data—i.e., an example $x \in \mathcal{X}$ is a sequence $x = (w_1, ..., w_m) \in \mathcal{W}^m$.[1] For example, each *subcomponent* $w$ may be a line of a poem or a measure of music, and $x$ may be a poem or song.

We are interested in domains where likely examples satisfy latent relational constraints $c \in \mathcal{C}$ over the subcomponents. For instance, $c$ may say that two measures $w_i$ and $w_j$ of $x$ start with the same series of pitches, or that two lines $w_i$ and $w_j$ of $x$ rhyme. We assume given a set of relations $\mathcal{R}$ (e.g., $r \in \mathcal{R}$ might be "rhyme" or "equal"), and a function $f : \mathcal{W} \times \mathcal{W} \times \mathcal{R} \to [0, 1]$ such that $f(w, w', r)$ indicates to what extent $w$ and $w'$ satisfy relation $r$. Then, $c$ is a compact representation of the relations present in an input $x$. We describe the structure of $c$ in detail in Section 3.1; for now, the approach we describe works for any choice of $c$. In particular, we build on *neurosymbolic generative models* (Young et al., 2019), where $c$ is itself generated based on a latent value $z \in \mathcal{Z}$—i.e.,

$$p_{\theta,\phi}(x) = \int \sum_{c \in \mathcal{C}} p_\theta(x \mid c) \cdot p_\phi(c \mid z) \cdot p(z) dz.$$

Then, Young et al. (2019) considers the variational distribution

$$q_{\tilde{\phi}}(c, z \mid x) = q_{\tilde{\phi}}(z \mid c) \cdot q(c \mid x) \qquad \text{and} \qquad q(c \mid x) = \delta(c - c_x).$$

Here, $\delta$ is the Dirac delta function and $c_x$ is a single representative generated from $x$ using a program synthesis algorithm (David & Kroening, 2017)—i.e., an algorithm $\mathcal{A}$ that takes as input an example

---

[1] We use a fixed $m$ to simplify our exposition; our approach trivially extends to variable $m$.

$x$ and outputs a program $c = \mathcal{A}(x)$ encoding the relational constraints present in $x$. Next, Young et al. (2019) derive an evidence lower bound

$$\log p_{\theta,\phi}(x) \geq \log p_\theta(x \mid c_x) + \mathbb{E}_{q_{\tilde{\phi}}(z \mid c_x)}[\log p_\phi(c_x \mid z)] - D_{\text{KL}}(q_{\tilde{\phi}}(z \mid c_x) \parallel p(z)) + H.$$

where $D_{\text{KL}}$ is the KL divergence and $H$ is information entropy. The first term is the log-likelihood of a generative model predicting the probability of example $x$ given relational constraints $c_x$, and the second and third terms form the loss of a variational autoencoder (VAE) $p_\phi(c \mid z)$ and $q_{\tilde{\phi}}(z \mid c)$ (Kingma & Welling, 2019). In summary, given training examples $x \in \mathcal{X}$, this approach separately learns (i) a VAE to generate $c$, trained on the relational constraints $c_x$ synthesized from each training example $x$, and (ii) a model to generate $x$ given $c_x$; the latter can take multiple forms such as a second VAE or a generative adversarial network (GAN) (Goodfellow et al., 2014). This approach is called *synthesis-guided generative models (SGM)* since it uses program synthesis to guide training.

To leverage this framework, we have to instantiate (i) the space of relational constraints $\mathcal{C}$, (ii) the synthesis algorithm $\mathcal{A} : \mathcal{X} \to \mathcal{C}$ used to extract a program encoding the structure of $x$, and (iii) the architectures of $p_\phi(c \mid z)$, $q_{\tilde{\phi}}(z \mid c)$, and $p_\theta(x \mid c)$. In prior work, Young et al. (2019) used heuristics specific to the the image domain to achieve these goals—in particular, they used (i) simple equality constraints on sub-regions of the image designed to capture 2D repeating patterns, (ii) a custom synthesis algorithm that greedily adds constraints in the data to the program, and (iii) a representation of $c_x$ as an image, in which case $p_\theta$ is a generative model over images, and $p_\phi, q_{\tilde{\phi}}$ based on an encoding of $c$ as a fixed-length vector.

We design a synthesis algorithm that expresses the synthesis problem as a constrained combinatorial optimization problem, which it solves using an SMT solver called Z3 (De Moura & Bjørner, 2008). In terms of (iii), our programs encode declarative constraints rather than imperative renderings, so the previous architectures of $p_\phi$, and $q_{\tilde{\phi}}$ cannot be used. Instead, we use expert domain-specific heuristics, transformers (Vaswani et al., 2017), or graph neural networks (GNNs) (Kipf & Welling, 2017) for $p_\phi$ and $q_{\tilde{\phi}}$. For $p_\theta$, we propose several methods for imposing the constraints encoded by $c$ when generating an example $x$.

## 3 Relational Constraints for Sequence Data

We describe how we represent relational constraints $r$, as well as our algorithm $\mathcal{A}$ for synthesizing the relational constraints $c_x = \mathcal{A}(x)$ present in an example sequence $x$.

### 3.1 Graph Representation of Relational Constraints

Recall that our generative model operates by first generating a relational program $c$, and then generating an example $x$ that satisfies $c$. Thus, for each training example $x$, we need to design a relational program $c$ that encode constraints on the structure of $x$. A program $c$ encodes a set of *relational constraints*, each of which imposes a constraint that subcomponents of $x$ should have certain kinds of relations. We begin by describing the structure of a single relational constraint, and then describe how $c$ encodes a set of relational constraints.

A *relational constraint* $\phi \in \Phi = \mathcal{W} \times \mathcal{I} \times \mathcal{R}$, where $\mathcal{I} = \{1, ..., m\}$, is a tuple $\phi = (\tilde{w}, i, r)$; we call $\tilde{w} \in \mathcal{W}$ a *prototype subcomponent*. An example $x$ *satisfies* $\phi$ to extent $h$ (denoted $x \models_h \phi$) if $f(\tilde{w}, w_i, r) = h$, where $w_i$ is the $i$th subcomponent of $x$. That is, $\phi$ says the $i$th subcomponent $w_i$ of $x$ should have relation $r$ with prototype subcomponent $\tilde{w}$. Thus, we can interpret $\phi$ as a function $\phi : \mathcal{X} \to [0, 1]$, where $\phi(x) = 1$ if $x$ satisfies $\phi$ to the maximal extent and $\phi(x) = 0$ if $x$ does not satisfy $\phi$ at all.

Next, a relational program $c_x$ encodes a set of relational constraints on examples $x$. We represent $c_x$ as an undirected labeled bipartite graph $c = (\tilde{V}, V, E)$ with vertices $\tilde{V}$ and $V$ and edges $E \subseteq \tilde{V} \times V \times 2^{\mathcal{R}}$, where $\mathcal{R}$ is the set of relations and $2^{\mathcal{R}}$ is the power set of $\mathcal{R}$. The vertices $\tilde{w} \in \tilde{V}$ are prototype subcomponents $\tilde{w} \in \mathcal{W}$; equivalently, they may be vector embeddings of prototype subcomponents. The vertices $i \in V = \{1, ..., m\}$ are the indices of subcomponents in $x$. The edges $e \in E$ are tuples $e = (\tilde{w}, i, R)$, where $R = [0, 1]^{|\mathcal{R}|}$. For tractability of synthesis, we impose the constraint that each $v \in V$ is part of a single edge $(\tilde{w}, v, R)$ (though $\tilde{v} \in \tilde{V}$ may be part of multiple edges). Finally, $c$

encodes the set of relational constraints

$$\Phi_c = \{(\tilde{w}, i, r, h) \mid (\tilde{w}, i, R) \in E \wedge R[r] = h\}.$$

In other words, $c$ includes the relational constraint that each subcomponent $w_i$ of $x$ should have all relations $r \in R$ with prototype $\tilde{w}$ to extent $h$, where $v$ is connected to $\tilde{w}$.

In this paper, for most examples, we consider binary relationships that have 0 or 1 as values, and informally state that a pair $\tilde{w}, i$ does not have a relationship $r$ if $f(\tilde{w}, i, r) = 0$. However, as we show in our experiments, non-boolean functions with values between 0 and 1 can be used as well.

For example, in Figure 1, the graph shown on the top right encodes a relational constraint $c_x$, and the top right shows an example $x$ that satisfies all the constraints $\phi \in \Phi_{c_x}$ with a value greater than 0. The nodes on the left-hand side of $c_x$ are prototype subcomponents $\tilde{w} \in \mathcal{W}$, each of which is a line of poetry. The nodes on the right-hand side correspond to indices $i$ (from $i = 1$ on top to $i = m = 10$ on the bottom); each one is labeled with a set of relations $R_i$. Then, $\Phi_{c_x}$ contains constraints $\phi = (\tilde{w}, i, R_i)$ for each edge $\tilde{w} \rightarrow i$ in the graph, which says that line $i$ of $x$ should have relations $r \in R_i$ with $\tilde{w}$. For instance, the last (10th) node in $c_x$ has constraints $R_{10} = \{\text{rhyme}, \text{meter}\}$, and is connected to prototype line $\tilde{w} =$"The harp that thou...". Thus, this edge encodes a constraint $\phi = (\tilde{w}, 10, R_{10})$ saying that the last line of $x$ should rhyme and have the same meter as $\tilde{w}$. Indeed, the last line of $x$ is $w_{10} =$"Who serve thee most...", which satisfies this constraint.

**Remark 3.1.** We use prototypes rather than direct relationships between components to ensure the size of the graph is tractable—with this choice, the graph is linear in the size of the input (assuming the number of prototypes is constant) rather than quadratic. A compact graph is both each to synthesize (for training) and train a model to generate (for generation). In our experiments, we show that our approach significantly outperforms attempting to generate full graphs (i.e., adjacency tensors).

**Remark 3.2.** We refer to $c$ as a program since it can be interpreted as a Datalog program (Ceri et al., 1989) (i.e., a relational logic program); in particular, $\Phi_c$ is a set of Datalog relations over $x \in \mathcal{X}$.

## 3.2 Synthesizing Relational Constraints

Recall that when training our generative model, we need to design a program synthesis algorithm $\mathcal{A}$ that synthesizes a relational program $c_x = \mathcal{A}(x)$ that best encodes the latent relational constraints present in each training example $x$. A key question is where the prototypes come from. We simply choose the prototypes $\tilde{w}$ to be actual subcomponents in $x$. Thus, $c_x$ encodes that subcomponents of $x$ are each related to one of a small number of distinguished subcomponents of $x$. As described below, we formulate the problem of synthesizing $c_x$ as a constrained optimization problem.

**Optimization variables.** The variables are a binary vector $H \in \mathbb{B}^m$ and a binary matrix $K \in \mathbb{B}^{m \times m}$. Intuitively, $H_i$ indicates whether subcomponent $w_i$ of $x$ is a prototype subcomponent in $c$, and $K_{ij}$ indicates whether $w_i$ is the prototype for subcomponent $w_j$.

**Constraints.** Our optimization problem has the following three constraints:

$$\psi_1 \equiv k_{\min} \leq \sum_{i=1}^{m} H_i \leq k_{\max}, \qquad \psi_2 \equiv \bigwedge_{j=1}^{m} \sum_{i=1}^{m} K_{ij} = 1, \qquad \psi_3 \equiv \bigwedge_{i=1}^{m} \sum_{j=1}^{m} K_{ij} \leq m \cdot H_i.$$

First, $\psi_1$ says that the number of prototype subcomponents is between $k_{\min}$ and $k_{\max}$. Next, $\psi_2$ says that every subcomponent $w_j$ corresponds to exactly one prototype subcomponent $w_i$. Finally, $\psi_3$ says that for every $i$, if $w_i$ is the prototype subcomponent of $w_j$ according to $K$, then it must be a prototype subcomponent according to $H$ as well.

**Objective.** The objective of our optimization problem is expressed in terms of a precomputed distance matrix $D \in \mathbb{R}^{m \times m}$, where $D_{ij}$ measures the similarity between components $w_i$ and $w_j$; smaller values indicate a greater degree of similarity. In particular, we define

$$D_{ij} = \frac{1}{\sum_{r \in \mathcal{R}} (f(w_i, w_j, r))},$$

i.e., $D_{ij}$ is the extent to which each $r \in \mathcal{R}$ are not satisfied by $w_i$ and $w_j$. Then, our objective (which is to be minimized) has the following three terms:

$$J_1 = \sum_{i,j=1}^{m} K_{ij} \cdot D_{ij}, \qquad J_2 = \sum_{i,j=1}^{m} \left( \sum_k K_{ki} \cdot K_{kj} \right) \cdot D_{ij}, \qquad J_3 = -\sum_{i,j=1}^{m} H_i \cdot H_j \cdot D_{ij}.$$

First, $J_1$ says that subcomponents should be similar to their prototypes. Second, $J_2$ says that subcomponents should also be similar to other subcomponents that share the same prototype. Third, $J_3$ says that different prototype subcomponents should be dissimilar.

**Optimization problem.** Our algorithm $\mathcal{A}$ uses Z3 to solve the optimization problem

$$(H^*, K^*) = \underset{H,K}{\arg\min} \{\lambda_1 \cdot J_1 + \lambda_2 \cdot J_2 + \lambda_3 \cdot J_3\} \quad \text{subj. to} \quad \psi_1 \wedge \psi_2 \wedge \psi_3, \qquad (1)$$

where $\lambda_1, \lambda_2, \lambda_3 \in \mathbb{R}_{\geq 0}$ are hyperparameters. Finally, to construct $c_x$, $\mathcal{A}$ chooses

$$\tilde{V} = \{w_i \mid H_i^* = 1\}, \qquad V = \{1, ..., m\}, \qquad E = \{(w_i, j, R_{ij}) \mid K_{ij}^* = 1\},$$

where $R_{ij} = \{r \in \mathcal{R} \mid f(w_i, w_j, r) = 1\}$—i.e., $\tilde{V}$ are the prototype subcomponents according to $H^*$, $E$ are the edges according to $K^*$, and $R_{ij}$ are the extent to which relations are satisfied by $w_i$ and $w_j$. Z3 is guaranteed to find the optimal solution; in the event that multiple such solutions exist, it chooses one nondeterministically. Intuitively, our approach should perform well when a handful of prototypes are sufficient to approximately capture the relational structure in the data, which appears to be true in the domains of rhyming poetry and melodies. Also, the user can define relations in a way that captures desired structure for any domain.

# 4 Relational Constraints in Neurosymbolic Generative Models

We describe our model for generating examples $x$. Recall that our approach proceeds in two steps: (i) generate $c$, and (ii) generate $x$ given $\Phi_c$. We describe each step in detail below.

## 4.1 Step 1: Generating Relational Constraints

The first step of our generative model is to generate relational constraints $\Phi_c$ using a VAE—i.e., $p_\phi(c \mid z)$ is a VAE with $p(z) = \mathcal{N}(z; 0, I)$ being a Gaussian distribution. The main choice is the architecture to use for the VAE. In particular, we consider a representation of $c$ as a sequence $(s_1, ..., s_m)$, where $s_i \in \{0, 1, ..., k\}$ for each $i$; intuitively, $s_i$ encodes that subcomponent $w_i$ should have the same prototype subcomponent as $w_{i-s_i}$, or if $s_i \leq 0$, that $w_i$ corresponds to a new prototype subcomponent. In practice, we found that in the music domain, the vast majority of examples could be described using $k \leq 6$, which decreased the number of possible values that could be predicted and simplified the problem; however, it would be possible in other domains for $k$ to be as large as $m - 1$.

More precisely, we initialize $\Phi_c = \varnothing$. Then, we generate the sequence $s_i \in \{0, 1, ..., k\}$ and $r_i \in \{0, 1, ..., m\}$ (where $r_i$ is represented as a binary vector of length $n = |\mathcal{R}|$) using either: (i) an LSTM-VAE, or (ii) a feedforward network whose output is iteratively sampled from as a categorical distribution and then used as input in the next step (see Appendix C.1 for details). For each $i$, we generate $(\tilde{w}, R_i)$ based on $s_i$ and $r_i$ according to the following approach: If $s_i = 0$, we generate a new prototype subcomponent $\tilde{w}$ using a domain-specific generative model, and add $\phi_i = (\tilde{w}_i, i, R_i)$ to $\Phi_c$. If $s_i > 0$, we let $\phi_i = (\tilde{w}_{i-s_i}, i, R_i)$.

## 4.2 Step 2: Generating Examples Given Relational Constraints

Next, we describe how we implement the second step $p_\theta(x \mid c)$ of our generative model. We propose three approaches for generating $x$ given $\Phi_c$; we give details in Appendix B.

**Approach 1: Constrained sampling.** We sample values $x \sim p_\theta(\cdot)$ by sequentially sampling $w_i \sim p_\theta(\cdot)$ from a pretrained generative model $p_\theta(w)$. We do so using rejection sampling at each step—i.e., we sample $w_i \sim p_\theta(\cdot)$ until we find $w_i$ satisfying $f(\tilde{w}, w_i, r) \approx h$ for each $(\tilde{w}, i, r, h) \in \Phi_c$. In addition, to speed up sampling, at each step of sampling $w_i$ (e.g., a word in a line or a pitch in a measure), we eliminate choices that violate $\Phi_c$.

**Approach 2: Constraint-aware embeddings.** We train a conditional variational autoencoder (cVAE) $p_\theta(w_1, ..., w_m \mid c)$ in the form of a graph convolutional network (GCN) that simultaneously generates all $m$ subcomponents in a way that satisfies $c$, and sample $x = (w_1, ..., w_m) \sim p_\theta(\cdot \mid c)$. The GCN takes as input embeddings of each prototype and subcomponent of $x$, and the adjacency matrix is given by the edges in $c$ (where the relation is encoded as an edge attribute). Then, the GCN-cVAE is trained using the standard VAE objective (Kingma & Welling, 2019), along with a semantic consistency loss that encourages the generated examples to satisfy $c$.

| Models | FD | GCN Disc. | RF |
|---|---|---|---|
| SGM (Ours) (A1) | 43.4 | 0.54 | 0.89 |
| SGM (Ours) (A2) | **32.7** | **0.63** | **0.79** |
| SGM (Ours) (A3) | 37.5 | 0.43 | 0.91 |
| SGM (Ours) (A2, No Synth. Ablation) | 42.1 | 0.42 | 0.89 |
| SGM (Ours) (A2, Greedy Synth. Ablation) | 40.5 | 0.50 | 0.88 |
| SGM (Ours) (A2, Continuous Relation) | 33.2 | 0.46 | 0.88 |
| Attention-RNN | 39.9 | 0.47 | 0.88 |
| MusicAutobot | 53.7 | 0.51 | 0.95 |
| StructureNet | 44.0 | 0.45 | 0.91 |

| Models | NLL | GCN Disc. | RF |
|---|---|---|---|
| SGM (Ours) (A2) | **1028.4** | **0.63** | **0.79** |
| MusicVAE | 1158.6 | 0.50 | 0.85 |
| MusicAutobot | 1760.0 | 0.51 | 0.95 |

Table 1: Results for the music domain. Left: We show negative log-likelihood ("NLL", lower is better) on the held-out human test set (i.e., by estimating the ELBo of the entire test set using sampling). Right: We show Fréchet distance on MusicVAE embeddings ("FD", lower is better). Both: We show the cross-entropy loss of the graph discriminator trained to distinguish synthesized programs of generated examples vs. held-out test set examples ("GCN Disc.", higher is better), and the accuracy of a random forest trained to do the same thing on a handcrafted featurization of the programs ("RF", lower is better). The highest score in each column is bolded. As can be seen, our approach with sampling strategy A2 outperforms the baselines on all metrics, also outperforming the ablation using the same strategy but without program synthesis (i.e., using the full adjacency tensor).

**Approach 3: Combinatorial optimization.** We sample $x \sim p_\theta(\cdot)$ by sequentially generating $w_i$ by solving an optimization problem whose objective is to maximize adherence to $\Phi_c$ plus additional terms encoding domain-specific heuristics encouraging $w_i$ to be realistic.

# 5  Experiments

We evaluate our approach on two domains: music and poetry generation. We provide details on experimental design and additional results in Appendix C.

## 5.1  Music Generation

We evaluated our approach on a music generation, where $x$ is a song and $w$ are measures of music. We consider 20 relations including equality, same rhythm, etc.; see Appendix C.2.

**Dataset.** We used songs from the Essen folk song corpus (Schaffrath, 1995), using 2223 for training and 555 for testing (after removing examples with less than 16 measures or that were not in the standard 4/4 meter). For this dataset, we used each of the three approaches A1, A2, and A3 described in Section 4 to sample $x \sim p_\theta(\cdot \mid c)$. For A1, we use a pretrained transformer called MusicAutoBot (Shaw, 2020). For A2, we require a generative model that constructs vector embeddings of measures; we use the pretrained version of Magenta's MusicVAE which embeds pairs of measures (Roberts et al., 2018) and adapted it to produce single-measure embeddings. We finetune all models on our training examples.

**Baselines.** We compare to MusicAutoBot, a pretrained and finetuned attention LSTM (Attention-RNN) (Waite, 2016), Magenta's 16-bar MusicVAE (pretrained and finetuned), and StructureNet, which integrates structure into an LSTM (Medeot et al., 2018). To show the importance of synthesis, we compare to an ablation that uses A2 but with full adjacency tensors instead of synthesizing compact representations of relational constraints, and one that uses a greedy synthesizer—i.e., at each step, greedily choose the single prototype and its relations that most increases (1). Finally, we consider using a continuous relation, namely, the cosine similarity of the MagentaVAE embeddings.

**Metrics.** We compare performance in terms of both high-level and low-level structure. For low-level structure, we use the negative log likelihood (NLL) on a held-out test set for MusicVAE, MusicAutobot, and our approach with strategy A2. The remaining approaches are not probabilistic (or estimating probabilities is intractable). For these approaches, we use a variant of the standard Fréchet distance (FD) score used to evaluate GANs (Borji, 2019)—i.e., the Fréchet distance between the MusicVAE (16-bar) embeddings of the generated music and the held-out test set.

For high-level structure, given a generated (or human) example $x$, we use our synthesis algorithm to synthesize its relational constraints $c_x = \mathcal{A}(x)$. Then, given a collection $C_{\text{gen}} = \{c_x \mid x \in X_{\text{gen}}\}$ of

| Models | GCN Disc. | FD |
|---|---|---|
| SGM (Ours, A1) | **0.69** | 21.5 |
| SGM (Ours, A3) | 0.62 | **13.51** |
| SGM (No Learned Structure Ablation) | 0.59 | 21.2 |
| GPT2 | 0.47 | 14.3 |
| GPT2-Opt | 0.56 | 14.4 |
| BERT | 0.50 | 54.9 |
| RichLyrics | 0.51 | 23.0 |

Table 2: Results for the poetry domain. We show Fréchet distance on SentenceBERT embeddings ("FD", lower is better), along with the cross-entropy loss of the graph discriminator trained to distinguish synthesized programs of generated examples vs. held-out test set examples ("GCN Disc.", higher is better). The best score in each column is bolded. As can be seen, our approach (SGM) with sampling strategy A1 outperforms all other approaches in terms of high-level structure, while our approach with sampling strategy A3 outperforms all baselines in high-level structure and is also competitive with GPT-2-based models in FD scores.

synthesized structure for generated examples, along with a collection $C_{\text{human}} = \{c_x \mid x \in X_{\text{human}}\}$ of synthesized structure for the held-out human examples, we train a graph convolutional neural network (GCN) to try and discriminate $C_{\text{gen}}$ from $C_{\text{human}}$, as well as a random forest (RF) over handcrafted features (see Appendix C.4). Intuitively, higher discriminative power should indicate less realistic structure. In both cases, we use a balanced dataset (i.e., 50% human held-out and 50% generated) so random predictions have accuracy 0.5. Recent work has shown that such discriminator-based metrics are valid for evaluating quality of generated examples (Lopez-Paz & Oquab, 2016).

**Results.** In Table 1, we show results for models for which we can compute the test set NLL (left) and results for the remaining models (right). As can be seen, our approach (SGM) with sampling strategy A2 outperforms all other models in both tables, in terms of both high-level structure and low-level structure. In Table 1 (left), the closest alternative is MusicVAE, for which the NLL is not too much larger; however, it performs significantly worse than our approach in terms of high-level structure. In Table 1 (right), we find that our other approaches also perform well (though not as well as A2). In particular, A1 performs well in terms of low-level structure, but is more mixed in terms of high-level structure. In contrast, A3 performs well in terms of high-level structure, but is mixed in terms of low-level structure, most likely since it does not use a learning-based model to generate low-level structure. Our ablation where we perform no synthesis performs poorly, especially in terms of structure, as does the one using greedy synthesis, demonstrating the importance of using constraint solving to synthesize compact representations of structure. On the other hand, greedy synthesis can be significantly more scalable than constrained optimization for large examples; thus, improving this strategy is an interesting direction for future work. Finally, using a continuous relation performs competitively in terms of FD score (though interestingly, it performs worse in terms of high-level structure), demonstrating that our approach can be applied with continuous relations.

## 5.2 Poetry Generation

Next, we apply our approach (SGM) to poetry generation; in this case, $x$ is a poem, and $w$ is a line. We consider two relations, rhyming and equal meter; see Appendix C.3 for details.

**Dataset.** We use from Project Gutenberg's poetry collection (Parrish, 2018), focusing on 10-line poems with rhymes and meter, with 2700 for training and 300 for testing.

**Our approach.** In the rhyming domain, due to the lack of rhyme-aware line embeddings, we did not perform A2. In applying A1, rather than sample words going forward, we sample them backwards, making it easier to sample lines that satisfy rhyming constraints; see Appendix B. Thus, we use BERT to sample (Devlin et al., 2018), since it is bidirectional. We apply A3 by performing constrained optimization to satisfy as many relations as possible while maintaining a low NLL.

**Baselines.** We compare to generation using beam search for BERT and GPT2 (Radford et al., 2019; Vaswani et al., 2017), both finetuned on our dataset. We also consider a variant GPT2-Opt of GPT2 where we use beam search to choose line breaks in a way that maximizes occurrences of rhyme and meter. We also tried a variant of GPT2 that used constrained sampling to try and find poems that fit a given rhyme and meter, but the search space was too large and it failed to generate a single poem even after several hours. We also compare to an implementation of RichLyrics (Castro & Attarian,

```
One was done. Another was done.          One was done. Another was done.
And I wish you know the way,              And I wish you know the way,
Full name and date to whom this story pour    Full name and date to whom this story pour
And know a lot of things that were called a war    And know a lot of things that were called a war
See a soldier, fair fair beautiful grace      See a soldier, fair fair beautiful grace
That men turn'd toward. Another race          That men turn'd toward. Another race
Together, married. Much to see, the dead      Together, married. Much to see, the dead
Were gone. The man who ascended to the head   Were gone. The man who ascended to the head
Office retired, and gave birth to a trace     With full beard and hair was a little said
That doesn't tell a name, but tells a face.   But was old and not intended for bed.
```

Figure 2: Left: Poetry generated using relational constraints $c \sim p_\phi(\cdot)$. Right: user modified variant of $c$ where the last two lines share a prototype with the two lines before them.

| Method | Average Score | Lyricism | Coherence | Rhyme/Meter |
|---|---|---|---|---|
| SGM (Ours, A1) | **3.66** | **3.81** | 3.59 | **3.59** |
| GPT2-Finetune | 3.30 | 2.90 | **3.91** | 3.12 |
| BERT-Finetune | 2.28 | 2.11 | 2.00 | 2.77 |
| RichLyrics | 3.09 | 3.24 | 3.09 | 2.93 |

Table 3: A user study evaluation in the poetry domain. While GPT2-Finetune outperforms our model in terms of coherence (presumably due to the well-known superiority of GPT-2 over BERT for generation), our method outperforms in terms of overall lyricism (i.e., whether the poem reads like poetry or prose), prominence of rhythmic/metrical structure, and average score.

2018), where the consecutive parts of speech for each line given the previous line and the ability to fill in the correct word for the given part of speech were both learned separately from the corpus. Finally, to show the importance of learning the distribution over constraints, we consider an ablation that uses A1, but sampling $\Phi_c$ uniformly randomly rather than from a learned distribution.

**Metrics.** For low-level structure, we use FD score on SentenceBert embeddings, which are unaware of rhyme and meter (Reimers & Gurevych, 2019); we cannot evaluate log-likelihood since we are using constrained sampling. For high-level structure, we train a GCN to discriminate synthesized programs for generated examples vs, test examples.

**Results.** We show results in Table 2. Our approach (SGM) with sampling strategy A3 significantly outperforms all baselines in terms of high-level programmatic structure, while also outperforming them in terms of FD scores. Approach A1 performs even better in terms of programmatic structure, but is not competitive with respect to FD scores, presumably due to the fact that GPT-2 is significantly better at natural language generation than BERT.

**User study.** We also performed a user study, discussed in Appendix C.5, which further confirmed this methods' strength in the poetry domain. in this domain, with 50 participants.

**User modifications.** A key benefit of our approach is that the user can modify the relational constraints $c$ (or construct their own from scratch) for use in the second step $p_\theta(x \mid c)$, giving the user a way to guide the generative process. An example in the poetry domain is shown in Figure 5.2, and musical examples are shown in Appendix 3.

# 6   Conclusion

We have presented a novel approach for representing and synthesizing relational constraints on sequence data, and for generating examples whose relational structure resembles that of the training data. Our experiments demonstrate that we outperform existing approaches in terms of achieving human-like structure, while performing comparably or better on both a user study and widely-used quantitative metrics that do not explicitly account for structure. Finally, our approach enables users to guide the generative process by modifying constraints. A key direction for future work is to apply our approach to other applications such as dialog generation and summarization, which may require novel programmatic structure compared to the ones we study.

**Limitations & ethical considerations.** We discuss limitations and ethical concerns in Appendix F.

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
