# A   Additional Related Work

**Music and poetry generation.** Both early music generation and poetry generation approaches were rule-based (Ovans & Davison, 1992; Atanassova & Pulov, 2002) or used simple statistical models such as Markov models (Sandred et al., 2009; Cope, 1987; Suleiman et al., 2017) or probabilistic CFGs (Quick, 2016; Thompson, 2009). Recent work has used deep learning to generate music (Huang et al., 2019; OpenAI, 2019) and poetry (Liao et al., 2019a); our experiments show that these approaches have difficulty generating realistic high-level structure.

Music generation has been approached using many machine learning techniques, including convolutional neural networks (CNN's) (Yang et al., 2017a), graph convolutional networks (GCN's) (Jeong et al., 2019), recurrent neural networks (RNN's) (Sübakan & Smaragdis, 2017)), and transformers (Huang et al., 2019). In particular, we leverage attention-based RNN's (Keerti et al., 2020) and transformers for constrained sampling of program structure.

Poetry has been approached with a variety of techniques as well, including relying on finetuned transformers (an approach we extend) (Liao et al., 2019b), RNN-based conditional-VAEs (Yang et al., 2017b), RNN-based planning (Wang et al., 2016), and GANs with both transformer and LSTM backends (Saeed et al., 2019).

Approaches have incorporated structure into deep learning to generate music (Medeot et al., 2018) or poetry (Castro & Attarian, 2018), but they are domain specific; we find they do not perform at a human level on capturing global (and sometimes local) structure. Some approaches incorporate expert-provided constraints such as rhyme and meter to generate poetry (Lau et al., 2018); unlike our approach, they cannot automatically learn and generate these constraints from data.

**Constrained text generation.** There has been work on constraining language models to produce outputs that satisfy a given decision function, or that maximize a given scoring function (Li & Rush, 2020; Miao et al., 2018; Dathathri et al., 2019). In contrast, our work focuses on settings where constraints are generated, and furthermore the distribution over constraints must itself be learned.

**Relational constraints and neural models.** Several previous works have focused on learning datalog programs, one component of our project. Mei et al. (2020b) learn Datalog programs which represent certain point processes, and use this to predict future events. Similarly, Mei et al. (2020a) also learn Datalog programs, and is able to achieve great success in certain autoregressive domains; however, they do not frame their problem as a generative process to generate realistic human data from their output.

# B   Generating Examples Given Relational Constraints

## B.1   Approach 1: Constrained Sampling

In the music domain, we choose the pretrained generative model $p_\theta(w)$ to be a pretrained version of MusicAutoBot. To generate $x$, we sequentially sample each measure $w_i$ conditioned on all prior measures $w_1, ..., w_{i-1}$. Each measure is sampled by sequentially sampling a sequence of pitch-duration pairs until the total duration is 16 beats (i.e., the length of a measure). During sampling, we mask pitch-duration pairs that cannot satisfy $\Phi_c$ (i.e., we set their sampling probability to zero and rescale the remaining probabilities). For instance, if the "has similar interval" relation is supposed to hold between the the prototype measure and measure $i$, and we are sampling the second note of measure $i$, then we mask any pitch $k$ in measure $i$ such that

$$|(\text{pitch}_k - \text{pitch}_{k-1}) - (\widetilde{\text{pitch}}_k - \widetilde{\text{pitch}}_{k-1})| \geq 3,$$

where $\widetilde{\text{pitch}}_k$ is pitch $k$ in the prototype corresponding to $w_i$. In other words, we eliminate pitches that would cause sampling to violate this constraint.

In the the poetry domain, we finetune a pretrained BERT model on our dataset, by taking the pretrained models weights and then training the model on our dataset with a strong gradient weight decay. BERT has the ability to complete masked words in a sentence. We leverage this ability to sample lines that rhyme and have the same meter, which is a challenging task since such lines are a tiny fraction of the search space. We describe how we simultaneously handle rhyming and equal meter; the cases where only one of these two constraints has to hold are similar. Given a prototype $\tilde{w}$,

we work backwards—on each step $j$, we sample from BERT a word $word_k$ that has the same number of syllables as the corresponding word $\widetilde{word}_k$ in the prototype. More precisely, we feed BERT the sequence

$$\widetilde{word}_1, ..., \widetilde{word}_{k-1}, \text{MASK}, word_{k+1}, ...$$

and ask it to fill in the masked word, setting the probability of any word with different number of syllables as $\widetilde{word}_k$ to zero. Here, $word_k$ is sampled according to the probabilities output by BERT for filling in the mask in the sequence with a single word. In other words, we can sample from BERT since we are always feeding it a complete sentence with a single masked word to be re-sampled.

To avoid producing a line which is similar to the original line, we also set the probability of any word equal or too similar to the original word in terms of GloVe cosine similarity (Pennington et al., 2014) to zero, except for in the case of the last word, where we instead restrict to words that rhyme with $\widetilde{word}_k$. To increase diversity, we sample the remaining words twice—(i) backwards-to-forwards from word $k-1$ to word 1, where $k$ is the number of words, and (ii) we resample each of the $k-1$ words (i.e., except the last word) in a random order. We discard any lines which, according to BERT, after being sampled are determined to be too unlikely when preceded by the previously generated lines.

## B.2 Approach 2: Constraint-Aware Embeddings

This approach uses a graph convolutional network (GCN) conditional variational autoencoder (cVAE), or GCN-cVAE, which consists of a GCN encoder $q_\theta(z' \mid x, c_x)$ and a GCN decoder $p_\theta(x \mid z', c_x)$.

In more detail, we assume $x = (w_1, ..., w_m)$ is represented as a sequence of vectors $(u_1, ..., u_m)$—e.g., in the music domain, we use MusicVAE $p_\psi$ to encode $u \sim q_\psi(\cdot \mid w)$ or decode $w \sim p_\psi(\cdot \mid u)$. Then, the latent encoding $z'$ consists of an embedding vector for each subcomponent $u$ of $x$.

Next, $c_x$ is incorporated into each GCN by converting it into a tensor with dimensions $|\hat{w}| \times |\hat{w}| \times |R|$ used as the adjacency matrix of that GCN (the last dimension is the edge attribute). Intuitively, the edges in $x$ are relations in $c$ between subcomponents and prototypes.

This GCN-cVAE it is trained using the usual VAE objective (Kingma & Welling, 2019): (i) a KL divergence term encouraging the embeddings $z'$ to be Gaussian, and (ii) a reconstruction loss in terms of mean-squared error. We also include a semantic consistency loss to enforce the satisfaction of the constraints $\Phi_c$. In particular, we train a classifier $p_\alpha(u, u'; r)$ that predicts whether two subcomponents $u, u'$ satisfy relation $r$ (more precisely, when decoded by $p_\psi$). The model $p_\alpha$ is trained examples $(u, u', r)$ from the training data $x$. Then, we include the loss

$$\sum_{(\tilde{w}, i, r) \in \Phi_c} p_\alpha(\tilde{u}, u_i, r),$$

where $\tilde{u} \sim p_\psi(\cdot \mid \tilde{w})$ is the encoding of $\tilde{w}$, $u_i \sim p_\psi(\cdot \mid w_i)$ is the encoding of $w_i$, and $r$ is a relation.

For the music domain, we use a pretrained MusicVAE for $p_\psi$ and $q_\psi$; unlike the MusicVAE we use for evaluation, we finetune a model that decodes 1 measure of music from a 256-dimensional vector.

## B.3 Approach 3: Combinatorial Optimization

Given sampled program $c$, this approach attempts to generate values $x = (w_0, \ldots, w_m)$ such that $x \models \Phi_c$ by solving a system of constraints. However, when generated using a neural network, relational constraints $\phi \in \Phi_c$ are not always consistent with one another, so we convert the constraint $x \models \Phi_c$ into an objective—i.e.,

$$x = \arg\max_{x \in \mathcal{X}} \sum_{i=1}^{m} \sum_{r \in \mathcal{R}} \mathbb{1}(\mathcal{R}(\tilde{w}, w_i, n) \Leftrightarrow (\tilde{w}, i, n) \in \Phi_c).$$

The ability to encode this optimization problem as one that Z3 can solve depends on the domain and relations. For this approach to work, we may need to include additional, handcrafted terms in the objective that encourage the generated example $x$ is realistic.

For the music domain, the optimization variables are the optimal sequence of pitches and their durations. The objective function is a linear combination of the degree to which $x$ satisfies $c$, along with domain-specific heuristics—e.g., minimizing large jumps in pitch values (i.e., $|\text{pitch}_{k+1} - \text{pitch}_k| \geq 4$), not having any intervals of length 6 (i.e., $|\text{pitch}_{k+1} - \text{pitch}_k| = 6$) due to the unpleasant harmonic nature of that interval, and not having two consecutive jumps in pitch (i.e., $|\text{pitch}_{k+2} - \text{pitch}_{k+1}| \geq 5) \wedge |\text{pitch}_{k+1} - \text{pitch}_k| \geq 5$). These heuristics are based on standard concepts from music theory (Horton & Ritchey, 2000).

For the language domain, we use GPT-2 to sample a line except for the last word; then, the optimization variables are the last words in each line. This strategy optimizes the relations between each line and its prototype, while leveraging GPT-2 to maintain low NLL for the entire poem.

## C   Evaluation Details

### C.1   Experimental Setup

**Synthesizing programs.** The hyperparameters $\lambda_1$, $\lambda_2$, and $\lambda_3$ in the program synthesis task, as described in the main section of this paper, regulate the degree to which the optimization favors solutions which have high similarity between prototype and sequence measures, have high similarity between elements sharing a prototype, and have high difference between prototypes, respectively. Their values were different with respect to the two different domains. In the poetry domain, $\lambda_1 = 1$, $\lambda_2 = 10$, $\lambda_3 = 1$, $k_{\min} = 1$, and $k_{\max} = 6$; in the music domain, $\lambda_1 = 1$, $\lambda = 5$, $\lambda_3 = 1$, $k_{\min} = 3$, and $k_{\max} = 6$. These values were chosen by manually inspecting the quality of the programs produced for 10 examples in the training set.

**Generating $c$.** To generate $c$ in the poetry domain, we use an LSTM-VAE with 6 LSTM layers and a latent size of 50. This model is trained to reproduce a given sequence of $(s_i, r_i)$ pairs which are given as input, with an additional requirement that the distribution of their encodings should be roughly equivalent to a Gaussian normal distribution. In the music domain, while we experimented with using an LSTM-VAE, empirically we had more success using a feedforward 3-layer network which took the previous $n$ (usually $n = 6$) $(s_i, r_i)$ pairs, and outputted a distribution over the following pair. The learning rate used was 1.0 (arrived at by grid search), and included the KL and reconstruction loss.

Each $(s_i, r_i)$ pair is represented as a $(S + |R|)$-dimensional vector, where $S$ is the maximum distance between objects with the same prototype and $R$ is the set of relations.

**High-level structure.** We evaluate high-level structure by using our algorithm to synthesize the relational constraints in every generated example—i.e., $C_{\text{gen}} = \{\mathcal{A}(x) \mid x \in X_{\text{gen}}\}$, where $X_{\text{gen}}$ is the set of examples generated using a model. Similarly, we can construct $C_{\text{human}} = \{\mathcal{A}(x) \mid x \in X_{\text{human}}\}$, where $X_{\text{human}}$ is the set of human-created examples held-out from the training dataset. Then, we evaluate high-level structure by training a model to try to discriminate $C_{\text{gen}}$ from $C_{\text{human}}$; if the model achieves lower performance, then the quality of high-level structure is higher. A general approach is to train a graph neural network—in particular, a graph convolutional network (GCN)—to do so; this model takes as input the graph structure of relational constraints $c$, along with vector embeddings of the prototype subcomponents, and outputs whether $c \in C_{\text{gen}}$ or $c \in C_{\text{human}}$. We balance the data so it consists of 50% human data and 50% generated data. We report the cross-entropy (CE) loss; higher values correspond to better generative models. In the music domain, we additionally used a random forest (RF) trained on a manual featurization of $c$. We report the accuracy of the RF; lower values (i.e., closer to 50%) correspond to better generative models.

We carefully tuned the hyperparameters of both the GCN and RF discriminators (as well as the MusicVAE used to evaluate the NLL). The main relevant parameters for the neural networks are their learning rates, which we tune using grid search. For the random forest, we performed extensive feature engineering to ensure good performance; the features we constructed are in Appendix C.4.

### C.2   Musical Relations Used

The following are the relations $r \in \mathcal{R}$ used in the music domain:

1. Measures $i$ and $j$ have the same pitch classes.
2. Measures $i$ and $j$ have the same pitch class prefix.

3. Measures $i$ and $j$ have the same pitch class suffix.

4. Measures $i$ and $j$'s pitches have an edit distance of 1.

5. Measures $i$ and $j$ have approximately the same interval structure.

6. Measures $i$ and $j$ have the same interval prefix.

7. Measures $i$ and $j$ have the same interval suffix.

8. Measures $i$ and $j$ have the same note (pitch + duration) prefix.

9. Measures $i$ and $j$ have the same note (pitch + duration) suffix.

10. Measures $i$ and $j$ have the same rhythm.

11. Measures $i$ and $j$'s rhythm has an edit distance of $\leq 2$.

12. Either measure $i$'s onsets are a subset of measure $j$'s onsets, or measure $j$'s onsets are a subset of measure $i$'s onsets.

13. Measures $i$ and $j$ have the same rhythmic and melodic contour.

14. Measures $i$ and $j$ have the same rhythmic and melodic contour prefix.

15. Measures $i$ and $j$ have the same rhythmic and melodic contour suffix.

16. Either the first or second half of measures $i$ and $j$ are identical.

17. Either both or neither of measures $i$ and $j$ have leaps.

18. Measures $i$ and $j$ fit within the same diatonic scale.

19. Either both or neither of measures $i$ and $j$ have syncopation.

20. Either both or neither of measures $i$ and $j$ have consecutive notes shorter than an eighth note.

21. (Continuous) The cosine similarity between the Measure-VAE embeddings of measure $i$ and measure $j$.

## C.3  Poetry Relations Used

The following are the relations $r \in \mathcal{R}$ used in the poetry domain:

1. Lines $i$ and $j$ have the same end rhyme.

2. Lines $i$ and $j$ have the same meter.

## C.4  Random Forest Features

The following are the manually constructed features used in the random forest discriminator for the music domain:

1. Mean number of relations between prototype and sequence measures.

2. Variance of number of relations between prototype and sequence measures.

3. Variance in histogram of prototype measure mappings.

4. Longest sequence $i \ldots j$ such that $w_i \ldots w_j$ all have the same prototype measure.

5. Number of pairs $(i, j)$ such that $\tilde{w}_i = \tilde{w}_j$ and $\tilde{w}_{i+1} = \tilde{w}_{j+1}$.

6. Mean distance between two measures with the same prototype.

7. Variance in distance between two measures with the same prototype.

## C.5  User Study Details

50 participants completed our study on Mechanical Turk. Each was paid \$5 to complete a survey with 12 questions (3 poems each from four sources, Ours, GPT2-Finetune, BERT-Finetune, and RichLyrics). All poems were chosen automatically by taking the top 3 examples from the generated datasets according to GPT2-log-likelihood. The participants were asked to rank the following 3 statements from "strongly disagree" to "strongly agree" (1-5) as follows:

| Method | Average Score | Lyricism | Coherence | Rhyme/Meter |
|--------|--------------|----------|-----------|-------------|
| SGM (Ours) | **3.66 ± 0.05** | **3.81 ± 0.11** | 3.59 ± 0.12 | **3.59 ± 0.07** |
| GPT2-Finetune | 3.30 ± 0.06 | 2.90 ± 0.11 | **3.91 ± 0.11** | 3.12 ± 0.07 |
| BERT-Finetune | 2.28 ± 0.05 | 2.11 ± 0.11 | 2.00 ± 0.11 | 2.77 ± 0.14 |
| RichLyrics | 3.09 ± 0.07 | 3.24 ± 0.10 | 3.09 ± 0.11 | 2.93 ± 0.12 |

Table 4: A user study evaluation in the poetry domain; we show means and standard errors. While GPT2-Finetune outperforms our model in terms of coherence (likely because GPT-2 outperforms BERT at generation), our method outperforms in terms of overall lyricism (i.e., whether the poem reads like poetry or prose), prominence of rhythmic/metrical structure, ad average score.

1. It is obvious that this is a poem
2. This text is coherent
3. I notice that this text has rhyme and meter

We show the results in Table 4; note that each score in the table is based on 150 labels.

## D Additional Results

### D.1 Comparison to Constraint Solving

We also considered a comparison to a constraint-based implementation called Motifate, with explicit attention to development of musical material (Muhammad Faisal, 2017). This approach was designed with heuristics for 3-beat measures, while our evaluation models anticipated 4-beat measures, so we could not obtain FD scores. Nevertheless, we found that even the structure was insufficient—its RF discriminator had accuracy 0.91, and its GCN discriminator had cross entropy loss 0.43, both of which are significantly worse than the other approaches.

### D.2 Conditioning on User-Provided Structures

Here we show how user modifications can occur in the music and poetry settings. By explicitly modifying $c$, we are able to generate two pieces of poetry or two tunes with similar internal patterns but with different structural characteristics.

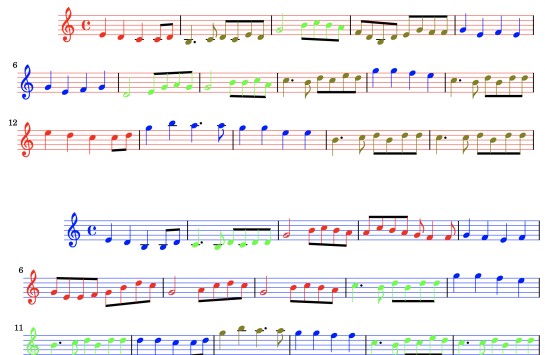

Figure 3: A song generated using our approach, and a nearly identical song generated where part of the sampled relational constraints $c$ were manually modified. These pieces were generated using A3 (constrained optimization), and the same reference measures $\tilde{w}$ were used, but $\Phi_c$ was slightly perturbed (the similarity relations were changed).

Furthermore, when we use constrained optimization (approach A3) to generate the music as well, we can hold constant the structure while imposing additional constraints that affect other musical aspects of the piece, such as harmony; see Figure 5.

One was done. Another was done.
And I wish you know the way,
Full name and date to whom this story pour
And know a lot of things that were called a war
See a soldier, fair fair beautiful grace
That men turn'd toward. Another race
Together, married. Much to see, the dead
Were gone. The man who ascended to the head
Office retired, and gave birth to a trace
That doesn't tell a name, but tells a face.

One was done. Another was done.
And I wish you know the way,
Full name and date to whom this story pour
And know a lot of things that were called a war
See a soldier, fair fair beautiful grace
That men turn'd toward. Another race
Together, married. Much to see, the dead
Were gone. The man who ascended to the head
With full beard and hair was a little said
But was old and not intended for bed.

Figure 4: Left: Poetry generated using relational constraints $c \sim p_\phi(\cdot)$. Right: User modified variant of $c$ where the last two lines share a prototype with the two lines before them.

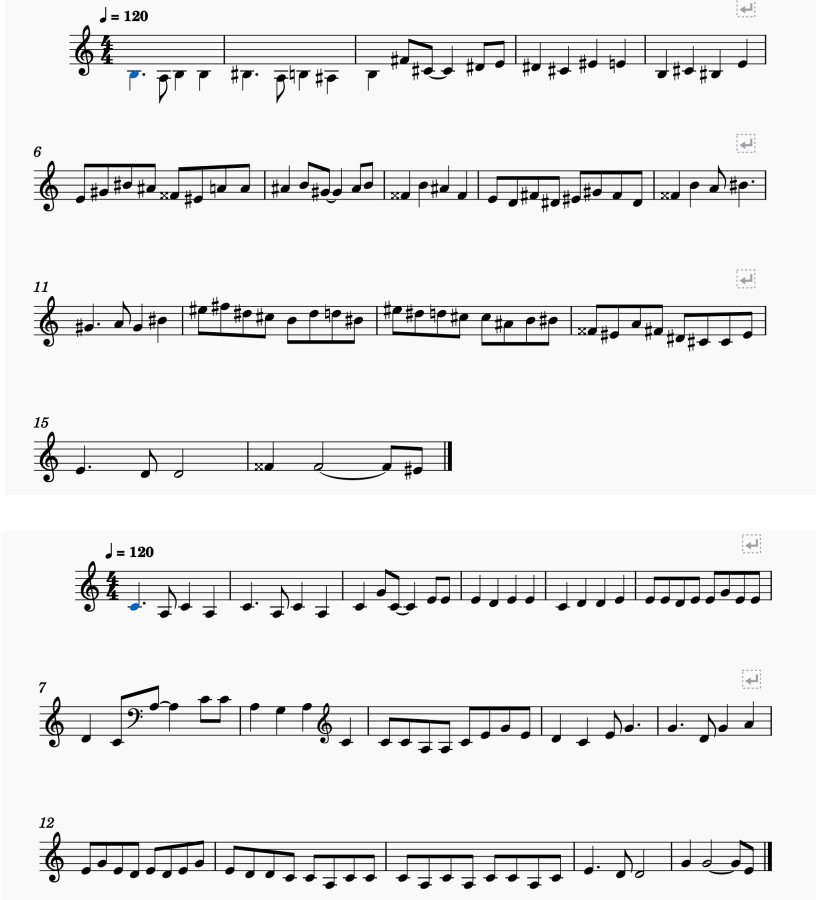

Figure 5: A song generated using our approach, and a song generated using the same prototype measures and relational constraints, but different harmonic qualities (chromatic vs pentatonic).

# E  Regarding comparison of melody-based music models to polyphonic/multi-instrument models

Our model produced coherent monophonic (one note at a time) melodies. In the past few years, large language models accompanied with datasets of over 200,000 pieces of polyphonic music have made generation of such music feasible, and the results often appear to be coherent. We focused on the monophonic domain since (i) we could contribute a discriminator for structure in the monophonic domain which could be extended to the polyphonic domain in future work, and (ii) because in doing so, we focused on one of the most promising areas for neurosymbolic research—i.e., the low-data

domain. With only 2700 training examples, one of these large language models (musicautobot) which has demonstrated efficacy in the large-data domain still underperformed related to our model.

### E.1 Algorithm Running Time

**Synthesis time.** Our algorithm uses the Z3 solver (De Moura & Bjørner, 2008) for synthesis. This solver uses a worst-case exponential time algorithm to ensure completeness (i.e., given sufficient compute time, it can always find the optimal answer), which is necessary given that the constraint solving problem is in general NP complete. In practice, Z3 is very efficient; in the poetry domain, all examples take less than 120 seconds to synthesize (without any special tuning), and often take much less time—e.g., only 3% took more than 10 seconds. In the music domain, while there were a handful of instances where the model failed to produce the provably optimal solution in the 120 seconds allotted, in all cases it produced a solution very close to the optimal one (by comparing its objective value to the optimal program obtained by running for longer than 120 seconds). Overall, the program synthesis component of training took a comparable amount of time to network training.

Our generation algorithms in Section 4.2 are generally very efficient. The constrained sampling strategy also worked well in the poetry domain due to the local nature of our constraints (for instance, rhyming effectively constrains only a single word). In our experiments, to find a valid sample, only 6.4 samples were needed on average and fewer than 15 samples for 95% of examples. Approach 3 (combinatorial optimization) runs similarly to the program synthesis algorithm described above. For general problems, Approach 2 (constraint-aware embeddings) always works well since it does not require any additional overhead during inference.

## F Limitations & Ethical Considerations

**Limitations.** Our work assumes that the relational primitives are given; the program synthesis algorithm also introduces additional hyperparameters; we found these parameters easy to tune in our domains by manually inspecting the synthesized programs. However, both of these choices require domain expertise; a key direction for future work is automatically discovering these primitives and tuning the hyperparameters.

In addition, our approach requires that this domain knowledge can be expressed in our language. We note that the predicates are very general, enabling our constraint language to capture complex relationships between subcomponents. In addition, while we have used a fixed value for the number of subcomponents $m$, we note that this assumption can be relaxed simply by allowing for "dummy" constraints that play the role of padding the example.

Finally, our sampling strategy can introduce additional overhead over existing approaches. They were tractable in our domains; in general, sampling Approach 2 should always scale well since it does not require constraint solving. We give a detailed discussion in Appendix E.1.

**Ethical considerations.** Increasing automation of music can impact the livelihoods of artists. In addition, we find that occasionally the poetry generation algorithm would produce something potentially offensive. Also, broadly speaking, our approach inherits the risks associated with large language models, including the ability to generate spam or other harmful content.

In general, we do not believe it introduces any new ones (beyond the risks of improving performance of these models); in fact, we believe our approach makes it easier for users to construct models that avoid accidentally generate harmful content by imposing constraints on the generated output. In addition, we plan on monitoring current research into removing toxicity from the output of large language models, and will apply these methods as they develop.

## G Qualitative Analyses

### G.1 Qualitative Observations on the Music Domain

In addition to quantitative measurements, we evaluated the strengths and weaknesses of our approach using A2 (which was the best according to quantitative metrics). According to our observations, the strengths of A2 include clearer phrases with obvious resolutions, likely and plausibly repetitive

rhythms, intervals between notes which seemed plausible but not overly repetitive, and less variance in quality. However, the results were not very rhythmically diverse, and certain idiomatic patterns of resolutions of intervals between notes and at the end of phrases were not followed. Furthermore, AttentionRNN does better in terms of creating realistic chord progressions (we did not explicitly consider chord progressions in our model; doing so is a promising direction for future work). Finally, while global structure is much better than the baselines, examples still relatively infrequently had the full four-bar repetitions characteristic of much folk music.

## G.2    Examples from the Music Domain

We show an example of generated songs using our approach with each A1, A2, and A3 in Figure 6, Figure 7, and Figure 8, respectively, and show an example generated using each of the baselines MusicVAE16, AttentionRNN, MusicAutoBot, and StructureNet in Figures 9, 10, 11, & 12, respectively. Qualitatively, the generated music and poetry appears plausible, exhibiting realistic high-level structure without sacrificing low-level structure.

Figure 6: An example of a song generated using our approach (A1). Measures that have the same prototype are shown in the same color. Note the existence of repeating four-bar phrases, found commonly in folk songs.

Figure 7: An example of a song generated using our approach (A2). Measures that have the same prototype are shown in the same color. Note the existence of clear phrase endings marked by long notes or rests, particularly the recurring pattern of fast notes resolving into long notes.

Figure 8: An example of a song generated using our approach (A3). Measures that have the same prototype are shown in the same color. The existence of two-bar and three-bar phrases is apparent, but the close note and rhythm similarities among different prototypes weaken the overall clarity of the song's melody.

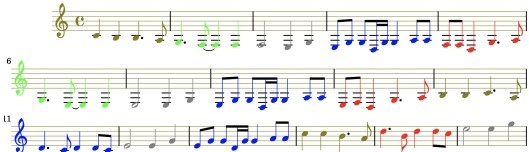

Figure 9: An example of a song generated using Magenta's hierarchical MusicVAE model finetuned on our dataset. While the local structure is extremely coherent, it does not seem to possess the expected internal repetition/development.

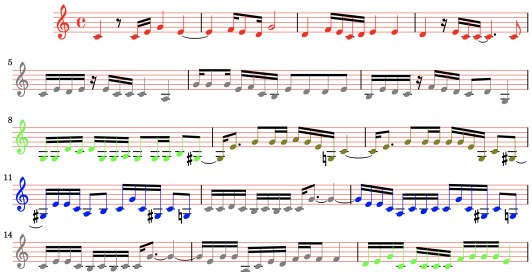

Figure 10: An example of a song generated using AttentionRNN trained on our dataset. Note the existence of erratic rhythms and unclear structure, which are common traits of custom-trained AttentionRNN models.

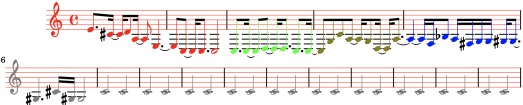

Figure 11: An example of a song generated using MusicAutoBot. Note the repetitive nature and stark contrast between the first half and second half of the song, which are common problems with transformer models.

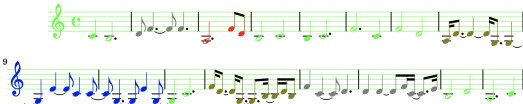

Figure 12: An example of a song generated using StructureNet. While some degree of internal structure is apparent, and the local coherence is high, the pattern of internal repetition seems fairly arbitrary.

## G.3 Examples from the Poetry Domain

In Figure G.3, we show an example poem generated using our approach (top) along with one generated using GPT2-Opt (bottom). As can be seen, the GPT2-Opt poem does not capture structure in the same way human poems do—e.g., adjacent lines are unrelated, lines have very unequal length, and the only rhymes are the word "the" in the brown lines and the words "to" and "too" in the green lines. There is even less structure in poems generated using vanilla GPT2. Thus, GPT2 is completely unable to capture high-level structure in the real poetry provided as training data. In contrast, our poem captures structure very similar to the human poem shown in Figure 1, such as rhyming adjacent lines.

We also give examples of poetry generated using our baselines—in particular, GPT2 finetuned and optimized for rhyme and meter in Figure 16, BERT finetuned as a language generation model in Figure 18, RichLyrics, and our ablation (i.e., use BERT in conjunction with a uniformly randomly sampled $\Phi_c$) in Figure 19.

One was done. Another was done.
And I wish you know the way,
Full name and date to whom this story pour
And know a lot of things that were called a war
See a soldier, fair fair beautiful grace
That men turn'd toward. Another race
Together, married. Much to see, the dead
Were gone. The man who ascended to the head
Office retired, and gave birth to a trace
That doesn't tell a name, but tells a face.

of nature and of Nature Nature Is the
only being able In human affairs to
combine with herself
Her will and therefore her existence Cannot ever fail Even
as nature having no desire can create itself so too
alone can Nature produce any being The
human existence cannot
then exist because
it only can
exist because the nature only is

Figure 13: Left: Poetry generated using relational constraints $c \sim p_\phi(\cdot)$. Right: Poetry generated by GPT2-Opt. Notice the lack of characteristic structure in GPT2-Opt, despite its coherence.

I know many things, and therefore I forgot,
Though I needed time to look ahead,
To understand something, time to let it fade away
As though it was yesterday as they
Were common things, free, rather—free, to go like the tide;
But another is to make no one, as it does.
Perhaps you know it. A queen, her beautiful son,
And another woman who has to go without one.
The voices like their cries of war,
They let us believe in a good restore!

Figure 14: An example of poetry generated using our approach (A1). Lines that have the same prototype are shown in the same color.

* your parents have both seen your face.
the sun in the west.
the moon under his robes coronation
the cloud that is the sky creation
the night that is the night.
the day that is the day station
you are friends and friends.
your smiles are fair and fair alteration
your friends are good and fair.
you have no shame to population

Figure 15: An example of poetry generated using our approach (A3).

Through the air and through the sky
And through all the world
I saw the sun the moon a star shine
In the midst of the stars
The stars were shining in my eyes my heart
Was throbbing with joy I felt
My heart was beating with love
I was
A little child in a little town
Where the little boys play

Figure 16: A poem generated using GPT2-Opt. It is more plausible than BERT in terms of global structure, which may be due to the fact that GPT2 is a better text generation tool than BERT, but it is still somewhat repetitive and its structure is not very human-like.

all all and and
and and and
and and all
all all all
and and o
and and and
o and and
o o and o and
and of of of and o o o but and and of
and and a and and last last last of of and and

Figure 17: A poem generated using BERT. It is clearly overly repetitive and not very semantically coherent, and lacks high-level structure.

and after all text that all more appointment
make its room from sat and all district without self
she one is hundred first enjoy her
but her been two you shall be one
above she leave enjoying the suffering usual
for which more science this day sewing
you shall two houses for recent contributions
and time which have left for self woman
and all moreover let use been found called
and might where out any boots not accident

Figure 18: A poem generated using RichLyrics. While it is less repetitive than non-conditioned BERT, it is still not very semantically coherent, and lacks high-level structure.

the first - independent , like for
the new songs for morning ,
a little world , they asked them for a way .
she asked them for a night ,
with two beds but sometimes lying on a light -
bed , the first for women , with another , one band ,
with one paul simon never got a play
on the subject , she ' d bought
a different dress for a different tent ,
and one dress for warning . .

Figure 19: A poem generated using our ablation. While it is much more coherent, it lacks the idiomatic rhyme and meter structure of our approach.