# OpenReview forum: "Neurosymbolic Deep Generative Models for Sequence Data with Relational Constraints"
_NeurIPS.cc/2022/Conference — NeurIPS 2022 Accept_

### Official Review · Reviewer_Jk4L · 2022-07-10

**Rating:** 5
**Confidence:** 3
**Soundness:** 4 excellent
**Presentation:** 2 fair
**Contribution:** 2 fair

**Summary:**

This paper studies a generative model for sequential data that has relational constraints. It proposes to use a two-stage generation process to (1) generate relation constraints (2) sample/generate the examples based on the relational constraints. Two domains, music and poem, have been used as evaluation data to test the correctness of the proposed method.

**Questions:**

Personally, I am hoping to see more concrete examples in the main paper to clearly show how the generated examples capture the relations with/without the proposed method.

**Limitations:**

I didn't find the section describing the limitation of this work other than the conclusion part, despite it is marked as 'Yes' in the checklist. Please correct me if I missed it.

**Strengths And Weaknesses:**

To me, this paper is on the borderline: on one hand, the topic and method presented are interesting and intuitive, while experimental results and analysis might be less convincing.

Pros:
(1) The topic and method are interesting and intuitive. Generating examples based on relational constraints seems to be a promising direction to consider for future work. This could further be extended to other domains.
(2) The author has spent efforts on user studies to examine the effectiveness of the method.

Concerns:
(1) The method still requires a lot of human effort in order to work well. I am not sure how easy it is to be used for new tasks.
(2) It appears to me that there is no golden metric to be used for evaluation in this specific context. The paper proposed at least two ways: the first one is based on NLL, which I am not sure if it really reflects the model is better in terms of capturing low-level structures; and the other is based on trained models, which I suspect if it can be biased for certain groups of methods.

---

> ### Author Response · Authors · 2022-08-02
> **Thanks for your feedback and a response**
>
> We thank you for your helpful feedback.  Regarding the points you brought up:
>
> **Comment:** The method still requires a lot of human effort in order to work well. I am not sure how easy it is to be used for new tasks.
>
> **Response:** First, we note that our approach can be easily applied in domains where the synthesizer has already been set up; e.g., a user can easily leverage their approach with their own music or poetry datasets. For a new domain, users need to invest more work; however, often very simple constraints can already lead to good performance. For instance, in the poetry domain, the straightforward rules we use are sufficient to substantially improve performance compared to our baselines. Finally, we strongly believe that future work will address these limitations; early deep learning approaches also required a lot of human effort to work well, and this effort has been reduced over time.
>
> **Comment:** Personally, I am hoping to see more concrete examples in the main paper to clearly show how the generated examples capture the relations with/without the proposed method.”
>
> **Response:** We provide additional examples in Appendix E; we will try to move these examples to the main paper if we have extra space, and are also happy to provide additional examples if needed.
>
> **Comment:** It appears to me that there is no golden metric to be used for evaluation in this specific context.
>
> **Response:** We share your concern that there does not exist a perfect evaluation metric for both global and local coherence, we believe that one of our contributions is to establish a rigorous metric for certain types of global coherence, and defer to the literature on existing methods of determining overall NLL.

---

### Official Review · Reviewer_aV2a · 2022-07-12

**Rating:** 8
**Confidence:** 4
**Soundness:** 3 good
**Presentation:** 4 excellent
**Contribution:** 4 excellent

**Summary:**

This paper proposes generative models for poetry and music which have natural relational constraints on their subcomponents -- metric/rhyme on lines for poetry, and repetition of measures. The approach is similar to a variational autoencoder where the continuous latent variable z generates a constraint sketch C, and the poem/music arrangement is generated based upon the sketch C. This sketch is defined with the assumptions that the final artifact has m subcomponents, and uses a subset of these subcomponents as prototypes to define additional relationships between the subcomponents and the prototypes. The sketch C|x is obtained deterministically by an SMT solver Z3, and rest of the model is trained to maximize the MLE via lower bound approximation. Empirical comparison is done with relevant prior work.

**Questions:**

-- How do you sample/generate from BERT since it is not a probabilistic model? Please describe the procedure for generation and discuss this in greater detail.
Relevant references:
BERT has a Mouth, and It Must Speak: BERT as a Markov Random Field Language Model, Wang and Cho
Exposing the Implicit Energy Networks behind Masked Language Models via Metropolis--Hastings, Goyal et al.

-- you mention “Z3 is guaranteed to find the optimal solution”. I am not sure how is it guaranteed. On the surface this looks like a NP-hard problem but their might be simplifying assumptions I may have missed.

**Strengths And Weaknesses:**

The paper tackles an important problem of incorporating structural constraints in otherwise opaque neural generative models and provides a clean solution to the domains explored in the paper. This kind of research is welcome in order to build more controllable and interpretable generative models. However, I feel the title/conclusion is slightly overclaiming because this approach I believe is limited only to a subset of relational constraints that are relevant to the domains explored in this paper.

The comparison is done with relevant baseline which clearly shows the effectiveness of the proposed approach over existing work.

However as mentioned earlier, this approach seems like it would work for very specific domains, where there is a clear mapping between prototypes and subcomponents and constraints are only defined on the subcomponents (Moreover, the constraints/algo in this paper only considers constraints over pairs of subcomponents). I am not sure how other kinds of constraints can be tackled in general. For example, even making the number of subcomponents (variable m) variable is not trivial practically, so a discussion about this aspect would improve the paper.

---

> ### Author Response · Authors · 2022-08-02
> **Thanks for your feedback and a response**
>
> We thank you for your helpful feedback.  Regarding the points you brought up:
>
> **Comment:** I am not sure how other kinds of constraints can be tackled in general.
>
> **Response:** First, we note that $m$ can be modified simply by allowing for “dummy” constraints that play the role of padding the example. More broadly, by encoding the appropriate predicates, our approach can capture complex relationships between different subcomponents. We will add a discussion to our paper.
>
> **Comment:** How do you sample/generate from BERT since it is not a probabilistic model? Please describe the procedure for generation and discuss this in greater detail. Relevant references: BERT has a Mouth, and It Must Speak: BERT as a Markov Random Field Language Model, Wang and Cho Exposing the Implicit Energy Networks behind Masked Language Models via Metropolis--Hastings, Goyal et al.
>
> **Response:** We employ BERT as a masked language model. We start with a random example of an individual line (e.g., generated by GPT2); then, for each word, we replace that word with one that has low NLL according to BERT while satisfying the constraint. We will clarify this point in our paper.
>
> **Comment:** You mention “Z3 is guaranteed to find the optimal solution”. I am not sure how is it guaranteed. On the surface this looks like a NP-hard problem but there might be simplifying assumptions I may have missed.
>
> **Response:** Z3 uses a worst-case exponential time algorithm. Thus, Z3 has the property that, given sufficient compute time, it can always find the optimal answer. In practice, Z3 is very efficient; each example takes less than 120 seconds to synthesize in our domain (without any special tuning), often taking much less time; for example, in the poetry domain, only 3% took more than 10 seconds.  In the music domain, while there were several instances where the model failed to produce a provably optimal solution in the 120 seconds allotted, in all cases it seemed generally able to produce nearly optimal solutions in that amount of time (as determined by choosing examples where it failed to provably optimize in 120 seconds, running them without the 120 second restriction, and then comparing rewards).  Overall, the program synthesis component of training took a comparable amount of time to network training. In general, these solvers have been tuned for many years and are highly efficient at practical problems such as the ones we need to solve.

---

> > ### Comment · Reviewer_aV2a · 2022-08-10
> > **Thanks for your response**
> >
> > Specifically, I found the description of Z3 helpful for my understanding.

---

### Official Review · Reviewer_kJEL · 2022-07-12

**Rating:** 7
**Confidence:** 4
**Soundness:** 3 good
**Presentation:** 3 good
**Contribution:** 3 good

**Summary:**

This paper proposes a method for generating sequences with relational structure. Specifically, the authors study the generation of poetry and music, where different lines of a poem have rhyme or equal meter constraints, and different measures of music have rhythm or pitch class constraints, etc. They represent these relational constraints as a bipartite graph, where the vertices on the left are the prototype subcomponents (i.e., lines or measures), the vertices on the right are the subcomponents to be generated, and the edge labels indicate the relations that the right endpoints need to satisfy with respect to the left endpoints.

To extract the relational constraints of a training example, the authors formulate it as a combinatorial optimization problem and use Z3 to solve it. To generate a sequence, they first use a VAE to generate relational constraints, and then generate subcomponents that satisfy these constraints, for which the authors provide three approaches: rejection sampling, conditional VAE, and combinatorial optimization.

Experiments on music and poetry generation show that the proposed method outperforms previous approaches in capturing relational structure.

**Questions:**

In more general, free-form essays, there are also connections between sentences or paragraphs. Can this method be extended to capture these structures to improve generation quality?

In line 196, for the objective $J_2$, should it be $\sum K_{ki}K_{kj}$ instead of $\prod K_{ki}K_{kj}$?

**Limitations:**

The proposed method seems to be suitable only for generating highly structured sequences with predefined relations. Also, the algorithm to extract the relational constraints has high time complexity.

**Strengths And Weaknesses:**

$+$ Despite doing well at generating locally coherent content, large language models still struggle to capture high-level structure. This paper considers relational structure between subcomponents and proposes a novel method to extract and learn to conform to them. This approach is particularly suitable for the generation of poetry and music with explicit relational constraints.

$+$ To implement the relational constraints, this paper makes solid technical contributions, including a program synthesis algorithm and a neurosymbolic generative model.

$-$ Experiments are only conducted on two small datasets with thousands of examples.

---

> ### Author Response · Authors · 2022-08-02
> **Thanks for your feedback and a response**
>
> **Comment:** In more general, free-form essays, there are also connections between sentences or paragraphs. Can this method be extended to capture these structures to improve generation quality?
>
> **Response:** Yes, we hope to look at this direction in the near future. Exciting potential research areas include collaborating with discourse theorists and combining sentence/paragraph embeddings with the more domain-driven insights of Rhetorical Structure Theory, as we experimented with both embedding-based similarities and domain-driven similarities.
>
> **Comment:** Should line 196 include a sum sign instead of a product sign?
>
> **Response:** Thank you for pointing this out; we will fix it.

---

### Official Review · Reviewer_Bid8 · 2022-07-15

**Rating:** 6
**Confidence:** 4
**Soundness:** 3 good
**Presentation:** 2 fair
**Contribution:** 3 good

**Summary:**

This paper proposes a deep generative model for sequence data, where the relational constraints between components in the sequence are modeled as latent variables. The overall framework is largely the same as Young et al. (2019) that is designed for image domain, while in the sequence data like text new challenges need to be addressed since the relationation constraints are different. The authors consider binary relations for the poetry and music domain, such as whether two lines of the poetry should rhyme. The model is learned with a VAE objective while being somewhat more complicated due to the symbolic relation variable. Specifically, the relational constraints need to be pre-extracted through solving a constrained optimization problem, then a separate VAE is learned on the relational constraints, and finally the authors propose three different approaches to model the data conditioned on the relation variable and the normal VAE latent variable. The model is overall interesting as a neurosymbolic deep model, which is claimed to model the global coherence better than typical sequence neural models and users could more or less control the generation through specifying the constraints. Experiments including human study confirm its superiority over normal NNs on generation such as MusicAutobot and GPT2.

**Questions:**

Questions: Is $f$ in Line 143 pre-known? I feel it is known from Eq. (1), right? If so, do you use prior knowledge/heuristics to obtain the paired relations in advance and compute $f$?

Suggestions: I think many details and the limitations should be included as mentioned in the weaknesses above, the presentation could also be improved.


**Limitations:**

The authors discussed the potential negative societal impact in the work but I don’t think the limitations were adequately discussed.


**Strengths And Weaknesses:**

### Strengths:

1. I think the authors make an attempt to address an important and difficult problem in sequence modeling by modeling symbolic relational constraints. Even though the method is not perfect (e.g. much more complicated than non-symbolic baselines), I would like to see such papers appear in the conference.
2. The method is novel (at least in the application to sequence data) and interesting. For example, extracting prototype subcomponents to obtain linear complexity for paired relations is clever, and solving it as a constrained optimization problem is a nice contribution.
3. Empirical results look strong, though I do have some concerns (see below).

### Weaknesses:

1. The method is much more complicated both methodologically and practically than the baselines, thus the empirical impact of the proposed method could be limited in reality – I guess not many people would use it practically. This is an important weakness that I have to mention here, yet I don’t think it is a sufficient reason to reject this paper since I think we should encourage this kind of work. The complexity of the method is expected since modeling symbolic variables in NNs is not easy. However, I think this paper should spare more space to discuss such complexity and limitations of this work. There are many limitations that need a discussion and many details are missing from the paper:
(1) There are many hyperparameters added by the method which the baselines do not have. How do you choose those hyperparams? How sensitive are the results to them? In fact, I cannot find the values of many hyperparams, e.g. $k_{min/max}, \lambda_{1/2/3}$.
(2) What is the overall computational cost and how does it compare to the baselines? How different are the efficiencies of the three approaches in Section 4.2? For example, the rejection sampling in Approach 1 may be slow. I think the efficiency aspect of the proposed method warrants a separate paragraph of discussion (at least in the appendix)
2. The paper is not well-presented. My overall feeling is that the authors try to compact too much content into the main body of the paper and therefore many details are missing and sometimes it could be difficult to follow. This is particularly true for Section 4 and 5.
3. Evaluation is important. Because of the lack of details as mentioned above, I am not sure whether the automatic evaluation in the paper is properly done. For example,
  (1) When you compute NLL, did you directly use one sample to estimate ELMO or use importance weighting to compute a tighter lower bound? The former is known to be inaccurate as a surrogate to NLL
  (2) When you train classifiers, what is the evidence to ensure that they are trained properly? What hyperparams did you try to train those classifiers? Did you try to tune them?
There is no need to tell too much on these details, but a paragraph in the appendix to convince readers that you did try to make evaluation convincing is much better.
4. I appreciate the user study evaluation, but I would like to know more details than the given ones in the appendix, for example, how many users score each example? If multiple users score the same example, do you have the agreement score for them (like variance)? – giving more details would make your results more convincing.

```
After author response:
The author response and the update of the paper addressed most of concerns, and I would like to increase my score to 6.
```

---

> ### Author Response · Authors · 2022-08-02
> **Thanks for your feedback and a response**
>
> Thank you for your helpful feedback. Regarding the points you brought up:
>
> **Comment:** The method is much more complicated both methodologically and practically than the baselines, thus the empirical impact of the proposed method could be limited in reality – I guess not many people would use it practically.
>
> **Response:** First, we note that our approach can be easily applied in domains where the synthesizer has already been set up; e.g., a user can easily leverage their approach with their own music or poetry datasets. For a new domain, users need to invest more work; however, often very simple constraints can already lead to good performance. For instance, in the poetry domain, the straightforward rules we use are sufficient to substantially improve performance compared to our baselines. Finally, we strongly believe that future work will address these limitations; early deep learning approaches also required a lot of human effort to work well, and this effort has been reduced over time.
>
> **Comment:** What is the overall computational cost and how does it compare to the baselines? For example, the rejection sampling in Approach 1 may be slow. I think the efficiency aspect of the proposed method warrants a separate paragraph of discussion (at least in the appendix)
>
> **Response:**	In our experiments, the constrained sampling was quite efficient; to find a valid sample, only 6.4 samples were needed on average and fewer than 15 samples for 95% of examples. This efficiency is due to the structure of constraints in our domain, which are fairly local. For general problems, Approach 2 always works well since it does not require any additional overhead during inference. We will add a discussion to our paper.
>
> **Comment:** When you compute NLL, did you directly use one sample to estimate ELMO or use importance weighting to compute a tighter lower bound? The former is known to be inaccurate as a surrogate to NLL
>
> **Response:** We computed NLL using the ELBo bound over the entire generated datasets (>2000 points each). Note that we use the ELBo across all approaches, so it is an apples-to-apples comparison.
>
> **Comment:** When you train classifiers, what is the evidence to ensure that they are trained properly? What hyperparams did you try to train those classifiers? Did you try to tune them?
>
> **Response:** The main relevant parameters for the GCN discriminators are their learning rates; we used grid search to optimize these parameters. We included the random forest (a very different kind of model) as a sanity check; for this model, we performed extensive feature engineering to ensure good performance. The features we used are provided in Appendix B.
>
> **Comment:** I appreciate the user study evaluation, but I would like to know more details than the given ones in the appendix, for example, how many users score each example? If multiple users score the same example, do you have the agreement score for them (like variance)? – giving more details would make your results more convincing.”
>
> **Response:**  50 participants scored each example, for a total of 150 labels per example. We will add standard errors to Table 4; these errors are small across all questions (the largest is 0.14).
>
> **Comment:** There are many hyperparameters added by the method which the baselines do not have. How do you choose those hyperparams? How sensitive are the results to them? In fact, I cannot find the values of many hyperparams.
>
> **Response:** We used grid search on a validation set to tune neural network learning rates. For the program synthesis hyperparameters (three hyperparameters to tradeoff different terms in the objective, and the kmin/kmax hyperparameters), we tuned them on a set of 10 training examples to optimize the match between the prediction and the actual example. We will add values of the hyperparameters to our paper (and also plan to open source our code).
>
> **Comment:** The authors discussed the potential negative societal impact in the work but I don’t think the limitations were adequately discussed.
>
> **Response:** We are happy to add a discussion of limitations as well as expand our discussion of the potential societal impact.
>
> **Comment:** Do you use prior knowledge/heuristics to obtain the paired relations in advance and compute f?
>
> **Response:** Our method works both in cases where there is domain knowledge to compute $f$ (such as in the case of determining whether lines rhyme) as well as in cases of automatically extracted knowledge such as measure embeddings, though in our experiments, the former (including domain knowledge) produced significantly better results.

---

> > ### Comment · Reviewer_Bid8 · 2022-08-07
> > **Thanks for your response!**
> >
> > Thank you for the response which addressed most of my concerns. The update of the paper is also appreciated. I still have one minor comment/suggestion:
> >
> >  > Note that we use the ELBo across all approaches, so it is an apples-to-apples comparison.
> >
> > My point was that ELBO is not a proper metric for evaluation since it often does not reflect the correct model rank in terms of NLL. Using a tighter lowerbound (https://arxiv.org/abs/1509.00519) for evaluation is more accurate.
> >
> > Anyway, I would like to increase my rating to 6 to reflect the author response.

---

> > > ### Author Response · Authors · 2022-08-07
> > > **Thanks!**
> > >
> > > Thank you for sharing the reference, we will take a look and do our best to incorporate it.

---

### Author Response · Authors · 2022-08-06
**Uploaded revision**

We thank the reviewers again for their helpful comments and feedback. We have updated our paper accordingly, with the following major modifications (highlighted in blue in the paper):

* Expanded discussion of BERT sampling (Appendix B.1)
* Added discussions of hyperparameter tuning (Appendix C.1)
* We have added user study details (Appendix C.5, Table 4)
* Added discussion of running time (Appendix D.6)
* Added a discussion of limitations and expanded discussion of societal impacts (Appendix E), referenced from the conclusion (Section 6); we are happy to move this section into the main paper if we can make space
* Added a discussion of expressiveness of the constraint language (Appendix E)

We have also updated our responses to individual reviewers to improve clarity.

---

### Meta-Review · Area_Chair_nUeK · 2022-08-26

**Recommendation:** Accept
**Confidence:** Certain

**Metareview:**

This paper proposes a deep generative model for sequence data by extracting relational constraints from data.  The model is overall interesting as a neurosymbolic deep model, which is claimed to model the global coherence better than typical sequence neural models and users could more or less control the generation through specifying the constraints. Experiments including human study confirm its superiority over normal NNs on generation such as MusicAutobot and GPT2.
All reviewers enjoy the novel neurosymbolic approach and appreciate the user feedback.

**Award:**

No

---

### Decision · Program_Chairs · 2022-09-14

Accept